# The Epidemiology of COVID-19 by Race/Ethnicity in Oklahoma City–County, Oklahoma (12 March 2020–31 May 2021)

**DOI:** 10.3390/ijerph19148571

**Published:** 2022-07-14

**Authors:** Kapil Khadka, Kunle Adesigbin, Jessica Beetch, Katrin Kuhn, Aaron Wendelboe

**Affiliations:** 1Epidemiology Division, Oklahoma City–County Health Department, 2700 NE 63rd Street, Oklahoma City, OK 73111, USA; kunle_adesigbin@occhd.org; 2Department of Biostatistics and Epidemiology, Hudson College of Public Health, University of Oklahoma Health Sciences Center, 801 NE 13th Street, Oklahoma City, OK 73104, USA; jessica-beetch@ouhsc.edu (J.B.); katrin-kuhn@ouhsc.edu (K.K.); aaron-wendelboe@ouhsc.edu (A.W.)

**Keywords:** SARS-CoV-2, public health surveillance, vulnerability, time-varying, public health response

## Abstract

We aimed to better understand the racially-/ethnically-specific COVID-19-related outcomes, with respect to time, to respond more effectively to emerging variants. Surveillance data from Oklahoma City–County (12 March 2020–31 May 2021) were used to summarize COVID-19 cases, hospitalizations, deaths, and COVID-19 vaccination status by racial/ethnic group and ZIP code. We estimated racially-/ethnically-specific daily hospitalization rates, the proportion of cases hospitalized, and disease odds ratios (OR) adjusting for sex, age, and the presence of at least one comorbidity. Hot spot analysis was performed using normalized values of cases, hospitalizations, and deaths generated from incidence rates per 100,000 population. During the study period, there were 103,030 confirmed cases, 3457 COVID-19-related hospitalizations, and 1500 COVID-19-related deaths. The daily 7-day average hospitalization rate for Hispanics peaked earlier than other groups and reached a maximum (3.0/100,000) in July 2020. The proportion of cases hospitalized by race/ethnicity was 6.09% among non-Hispanic Blacks, 5.48% among non-Hispanic Whites, 3.66% among Hispanics, 3.43% among American Indians, and 2.87% among Asian/Pacific Islanders. COVID-19 hot spots were identified in ZIP codes with minority communities. The Hispanic population experienced the first surge in COVID-19 cases and hospitalizations, while non-Hispanic Blacks ultimately bore the highest burden of COVID-19-related hospitalizations and deaths.

## 1. Introduction

Oklahoma comprises 1.2% of the United States (US) population [1], and the distribution of race and ethnicity in Oklahoma County nearly mirrors the US (Appendix A
Table A1). One notable difference is the relatively high proportion of Native American citizens. The first case of COVID-19 in Oklahoma was reported on 6 March 2020 in a Tulsa County resident who had travelled home from Italy [2]. Since then, the virus has been reported in every county in Oklahoma. Oklahoma County, the most populous county in the state, covers the Oklahoma City metropolis and surrounding larger towns. The first case in Oklahoma County was reported on 12 March 2020 [3]. During the study period, the county experienced 2 major surges of COVID-19 cases, hospitalizations, and deaths—the first during July–August 2020 and the second during November 2020–January 2021. The latter surge was the largest, with the 7-day average cases peaking at 850 cases (on 25 November 2020), 26 hospitalizations (on 8 January 2021), and 14 deaths (on 5 December 2020).

The COVID-19 vaccine was first available in Oklahoma County on 14 December 2020. As of early June 2021, Oklahoma had administered just under 3,000,000 COVID-19 vaccine doses and had fully vaccinated over 1,336,000 people (approximately 34% of the state) [4]. During the same period, Oklahoma County had fully vaccinated almost 263,000 people (33% of the county’s population). Those aged 12 years and above had a vaccination rate of 39.9%, while those aged 65 years and above had a vaccination rate of 66.9% [5]. Statewide, 26,489 people have been hospitalized due to COVID-19 [4], with 4978 (18.9%) of those occurring in Oklahoma County [5].

Many health disparities and factors affecting health are caused by racism [6,7]. These disparities have persisted through, and been exacerbated by, the COVID-19 pandemic [8,9]. For example, non-Hispanic Black and Hispanic people reported higher rates of COVID-19-related hospitalizations [8,9]. Another example is the uneven uptake of the COVID-19 vaccines by race and ethnicity [10]. Given the length of the COVID-19 pandemic and the differences in strategies that governments, communities, and individuals employed in response, it is uncertain the extent to which racial and ethnic communities were impacted by COVID-19 over time. Hence, our primary research question was whether vulnerable populations were equally vulnerable during the course of the pandemic, or if being impacted early bestowed protection against COVID-19-related outcomes later in the pandemic. Because social inequities penetrate virtually every aspect of life, we aimed to examine the epidemiological profile of reported cases, hospitalizations, and deaths from COVID-19 by race/ethnicity in Oklahoma County. This included examining the geographical hot spots of these outcomes by race/ethnicity. Further, we sought to examine the impact that vaccination had on these trends. Overall, the results of this report will contribute to a broader understanding of the epidemiology of COVID-19 in a county with relatively high vaccine hesitancy [11].

## 2. Materials and Methods

### 2.1. Study Design and Data Source

We conducted an observational study using population-based surveillance data. The source population was people living within the jurisdiction of the Oklahoma City–County Health Department (OCCHD). For the purpose of this study, Oklahoma City–County included all Oklahoma County residents as well as individuals living in Oklahoma City ZIP codes—including Oklahoma City ZIP codes that lie outside of Oklahoma County. The OCCHD jurisdiction (Appendix A
Figure A1) covers 56 ZIP codes as defined by the agency’s 2021 Wellness Score methodology [12]. Census data for 2019 were used to establish the population in this study [13]. Surveillance data—reported to the Oklahoma State Department of Health—of confirmed COVID-19 cases residing in Oklahoma City–County during 12 March 2020–31 May 2021 were included in this analysis. Vaccination data were obtained from the Oklahoma State Immunization Information System. Any data with missing ZIP code information were excluded from the analysis. In addition, any cases with unknown and/or non-reported race/ethnicity were excluded from any analysis involving racial/ethnic group.

### 2.2. Data Management

Race/ethnicity was categorized as non-Hispanic American Indian, non-Hispanic Asian/Pacific Islander, non-Hispanic Black, non-Hispanic White, and any person with Hispanic origin. The terms Hispanic and American Indian are used for the measure to accurately reflect those data captured in the surveillance. Hispanic is defined as those who are Spanish speakers and is inclusive of those who identify as Latinx. American Indian is defined as those who are members of any of the indigenous peoples of America. Age was categorized into groups of 0–4 years, 5–17 years, 18–35 years, 36–49 years, 50–64 years, and 65 years and older. Comorbidities included acute respiratory distress syndrome, chronic heart or circulatory disease, chronic liver disease, chronic lung disease, chronic renal failure, diabetes, disseminated intravascular coagulopathy, and pneumonia. We categorized the time period according to phases of the pandemic and observed changes in the hospitalization rate during those phases. The phases were defined as 12 March 2020–31 May 2020; 1 June 2020–31 October 2020; 1 November 2020–15 January 2021; and 16 January 2021–31 May 2021.

### 2.3. Data Analysis

The daily racial-/ethnic-specific hospitalization rates were calculated using the racial/ethnic population as the denominator. In addition, the proportion of cases that were hospitalized was calculated stratified by race/ethnicity. Pearson correlation coefficients were estimated when comparing racially-/ethnically-specific hospitalization rates to vaccination rates. The median and interquartile range (IQR) were calculated to describe the magnitude of hospitalization during different time periods of the pandemic. Logistic regression was used to estimate odds ratios (OR) and 95% confidence intervals (CI) to measure the association between race/ethnicity and being a case of, being hospitalized with, or dying from COVID-19; non-Hispanic White race served as the referent group. Sex, age, and the presence of at least one comorbidity were chosen a priori as factors for which to adjust. A type I error rate of 0.05 was used to test for statistical significance. Statistical analyses were conducted in the R Project for Statistical Computing.

Hot spot analysis was performed using normalized values of cases, hospitalizations, and deaths generated from incidence rates per capita (per 100,000 population). Environmental Systems Research Institute (Esri) geographic information systems software, ArcGIS Pro 2.8.2 (Esri, Redland, CA, USA), was used for the analysis.

## 3. Results

During the study period, there were 103,030 confirmed COVID-19 cases (incidence rate = 10,776 cases/100,000 population), 3457 COVID-19-related hospitalizations (hospitalization rate = 362/100,000 population), and 1500 COVID-19-related deaths (cause-specific mortality rate = 1.5%). In long-term care facilities, there were 1724 (1.7%) COVID-19 cases, 238 (6.9%) COVID-19-related hospitalizations, and 266 (17.8%) COVID-19-related deaths. The distribution of reported COVID-19-related infections and vaccination status among all residents are summarized in Table 1.

The distribution of COVID-19-related deaths, hospitalizations, demographic characteristics, and comorbidities among reported infections stratified by race/ethnicity is shown in Table 2.

There were more hospitalizations per capita among the non-Hispanic Black population and more deaths among the non-Hispanic White population. The presence of comorbidities appeared to be lowest for Asian/Pacific Islander and Hispanic populations. Hispanic COVID-19 cases tended to be in younger persons, while non-Hispanic White COVID-19 cases tended to be in older persons. Of people diagnosed with COVID-19, the Asian/Pacific Islander population in Oklahoma City–County had the lowest rate of comorbidities (9.2%), compared to 17.1% among the American Indian population and 17.0% among the non-Hispanic Black population. (Additional detailed information about the distribution of comorbidities by race/ethnicity is shown in Appendix A
Figure A2).

The 7-day average hospitalization rate per 100,000 population stratified by race/ethnicity peaked on 8 January 2021 (Figure 1). Early in the pandemic (June 2020 through mid-August 2020), the hospitalization rate was highest among the Hispanic population, but during the peak of the pandemic (December 2020–January 2021), the non-Hispanic Black population had the highest hospitalization rate. In contrast, the Asian/Pacific Islander population experienced the lowest hospitalization rates throughout the course of the pandemic.

Vaccination efforts began in mid-December 2020 in Oklahoma City–County. Health care providers were eligible for vaccination starting 14 December 2020, and people aged 65 years and older were eligible in early January 2021. The Asian/Pacific Islander population had the highest vaccination rate (45.6%), followed by the non-Hispanic White population (34.8%). Only about 20% of the eligible population (i.e., 5+ years) among the non-Hispanic Black population and 24% among the Hispanic population had been fully vaccinated by 31 May 2021.

Crude and adjusted odds of COVID-19-related infection, hospitalization, and death relative to race/ethnicity are summarized in Table 3. The American Indian, non-Hispanic Black, and Hispanic populations had significantly higher odds of being a case of COVID-19, as compared to the non-Hispanic White population. When adjusting for sex, age, and the presence of at least one comorbidity, non-Hispanic Black race was associated with hospitalization (OR, 1.4; 95% CI, 1.1–1.8) and death (OR, 1.7; 95% CI, 1.2–2.5) when compared to the non-Hispanic White population.

There is notable variation in hot spots of cases, hospitalizations, and deaths in Oklahoma City–County (Figure 2). Hot spots of cases were detected in contiguous ZIP codes in the southwest region of the study area (Figure 2a). This is a cluster of ZIP codes with considerably higher rates of cases per capita. The demographic breakdown shows Hispanic and non-Hispanic White populations accounted for the largest proportions of residents in this region (Figure 2c). On the other hand, hot spots of hospitalizations were detected in the central east region of the study area (Figure 2b). The non-Hispanic Black population made up the largest proportion of the population in this cluster. This region also produced hot spots of deaths (Figure 2b), and over 70% of residents in these hot spots were non-Hispanic Blacks (Figure 2d).

We found a decreasing hospitalization rate (7-day average per 100,000 population) with increasing vaccinated populations. The negative correlation between hospitalization rate and percent of people fully vaccinated was statistically significant (Pearson’s r = −0.77; 95% CI, −0.83, −0.70; *p*< 0.01). When stratified by race/ethnicity, the negative correlation between hospitalization rate (7-day average per 100,000 population) and percent of people fully vaccinated was statistically significant for all races/ethnicities (American Indian: Pearson’s r = −0.57; 95% CI, −0.67, −0.44; *p* < 0.01; Asian/Pacific Islander: r = −0.65; 95% CI, −0.73, −0.55; *p* < 0.01; non-Hispanic Black: r = −0.73; 95% CI, −0.80, −0.65; *p* < 0.01; Hispanic: r = −0.47; 95% CI, −0.59, −0.34; *p* < 0.01; non-Hispanic White: r = −0.78; 95% CI, −0.84, −0.71; *p* < 0.01). While the number of COVID-19 cases and hospitalizations peaked before the COVID-19 vaccine was available, the rate of hospitalization changed following the introduction of the vaccine. Specifically, there were a median of 2 [IQR, 1–4] COVID-19-related hospitalizations during 12 March 2020–31 May 2020; 6 [IQR, 4–9] hospitalizations during 1 June 2020–31 October 2020; 19 [IQR, 14–23] during 1 November 2020–15 January 2021; and 5 [IQR, 3–7] hospitalizations during 16 January 2021–31 May 2021.

## 4. Discussion

The COVID-19 pandemic impacted community members differently in Oklahoma City–County depending on race and ethnicity. American Indian and Hispanic populations experienced the highest burden of reported COVID-19 infections, and the non-Hispanic Black population had the highest adjusted odds of COVID-19-related hospitalizations and deaths. Because we were not able to link vaccination status to case status, we could not directly measure the protective benefit of being vaccinated against COVID-19. However, correlations are strong, given that the Asian/Pacific Islander population had the highest vaccination uptake and simultaneously accounted for the lowest proportion of cases, hospitalizations, and deaths. In contrast, the non-Hispanic Black population had the lowest vaccination uptake. An exception to this correlation is that the vaccination uptake among the Hispanic population was similar to the non-Hispanic Black population, but the Hispanic population experienced lower COVID-19-related death rates.

The COVID-19 pandemic also differentially impacted community members by race/ethnicity with respect to time. Hispanic community members were hospitalized with COVID-19 early in the pandemic, with the first surge occurring June 2020–August 2020. They also experienced a second peak in COVID-19-related hospitalization during the second surge occurring November 2020–January 2021, and they were the first to peak in early December 2020. In contrast, COVID-19-related hospitalizations peaked among the other races/ethnicities in late December 2020–early January 2021. While our findings align with those from other studies that report that the non-Hispanic Black population disproportionately experienced higher odds of hospitalization when compared to the non-Hispanic White population [14,15,16,17,18,19] (with the exception to the population in New York City) [20], our findings differ in that the Hispanic population did not have a higher burden of COVID-19-related hospitalization and death. Although we do not have data to inform the reason for this difference, it has been reported that the Hispanic population in Oklahoma County has a lower mortality rate than for non-Hispanic populations [21]. These findings are informative for clinical providers, researchers, and policymakers so that results from studies in one location are not overgeneralized to other populations that share the same racial/ethnic heritage.

The communities where hot spots of cases, hospitalizations, and deaths were detected in the study area have been established in previous analyses by the Oklahoma City–County Health Department as areas of public health concern [12]. Compared to the rest of the study area, these communities have experienced adverse health and socioeconomic outcomes, and the Centers for Disease Control and Prevention (CDC) has identified them to have more social vulnerabilities [22]. These hot spot communities also have higher mortality rates from chronic diseases, higher rates of poverty and unemployment, as well as lower income and educational achievement. Our findings from the COVID-19 hot spot analysis revealed that hot spots of cases occurred in ZIP codes with high proportions of Hispanic and non-Hispanic White populations, while hot spots of hospitalizations and deaths occurred in ZIP codes with a high proportion of the non-Hispanic Black population. A closer look at the demographic breakdown revealed that socioeconomic status varies significantly by race and ethnicity within these hot spots. For example, in ZIP codes where Hispanic or non-Hispanic Black populations account for at least 50% of the total population, the median household income is $32,227 and $39,059, respectively. This compares to a median household income of $74,488 in ZIP codes where the non-Hispanic White population accounts for at least 50% of the total population [13]. Poverty is higher and average education is lower in the ZIP codes where the majority of residents are Hispanic or non-Hispanic Black, as compared to ZIP codes with predominantly non-Hispanic White residents. These hot spot analyses underscore the need to empower residents in the identified hot spots by increasing health literacy and providing additional public health infrastructure to improve equity in overall health among minority and vulnerable populations.

The factors contributing to health disparities and structural racism are complex and have historical roots [23]. An example of these disparities having long-term impacts on health is the finding that Black Americans receive poorer quality of perinatal and neonatal care, both of which are associated with increased chronic conditions later in life [24]. There is also a lack of representation of minorities in the field of public health [25]. Many of these factors can be addressed within the local context. Place-based, multisector, equity-oriented initiatives have been suggested as a strategy to address structural racism [23]. An example of a place-based strategy currently being employed in Oklahoma County includes hiring and embedding community health workers in community-based organizations that serve minority populations.

Our findings are limited by the standard limitations of public health surveillance data, including delayed and under-reporting of data, differential access to care, and inadequate public health surveillance information technology. For example, the database in which vaccination records are maintained (Oklahoma State Immunization Information System) was not linked to the disease investigation and reporting system (Public Health Investigation and Disease Detection of Oklahoma). We included confirmed COVID-19-related deaths, as defined by the Oklahoma State Department of Health, which is fewer than the provisional COVID-19-related deaths reported by the CDC. In addition, these surveillance data do not include access to medical insurance, which likely impacts the hospitalization rates by race/ethnicity.

## 5. Conclusions

In summary, understanding the epidemiology of COVID-19 in Oklahoma City–County provides additional insights into racial-/ethnic-specific impacts of the timing of COVID-19-related hospitalizations, the burden of COVID-19-related deaths, and the COVID-19 vaccination uptake. By examining the geographic location by ZIP code, policymakers are better able to integrate previous efforts with current and future plans to increase the resiliency of populations who are most vulnerable to poor health outcomes. We recommend increased incorporation of representatives from minority populations in public health-related initiatives. This includes inviting active participation from community members who are impacted by these initiatives, as well as recruiting and training community members to work in public health.

## Figures and Tables

**Figure 1 ijerph-19-08571-f001:**
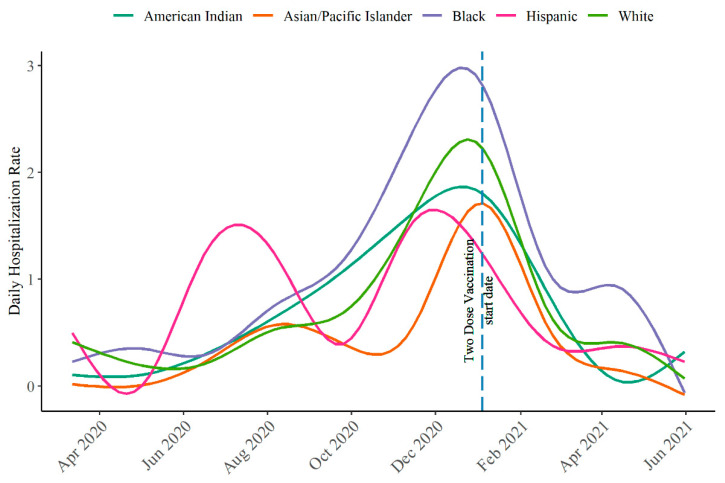
The 7-day average of daily hospitalization rate per 100,000 population smoothed trend lines stratified by race/ethnicity in Oklahoma City–County, 12 March 2020–31 May 2021. The vertical dashed line indicates the first date by which an Oklahoma City–County resident could have received two COVID-19 vaccine doses.

**Figure 2 ijerph-19-08571-f002:**
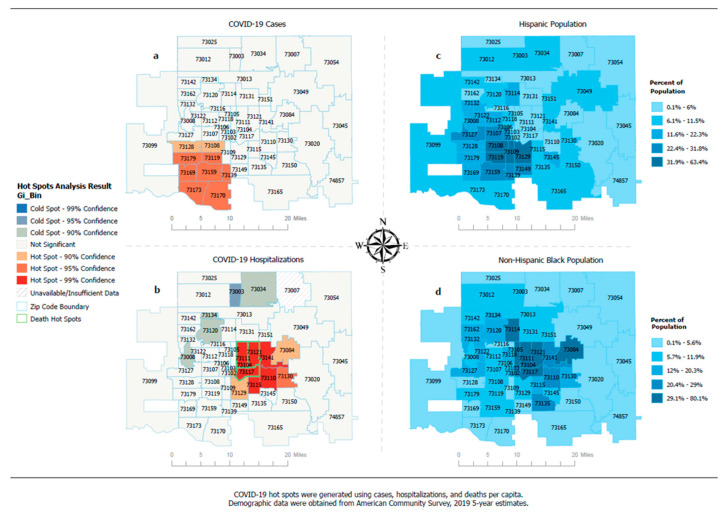
(**a**) Hot spots of COVID-19-related cases. (**b**) Hot spots of COVID-19-related hospitalizations and deaths. (**c**) Distribution of the Hispanic population in the Oklahoma City–County metropolitan area. (**d**) Distribution of the non-Hispanic Black population in the Oklahoma City–County metropolitan area.

**Table 1 ijerph-19-08571-t001:** The distribution of reported clinical characteristics by population-based race/ethnicity in Oklahoma City–County, 12 March 2020–31 May 2021. The denominator for vaccination status is the population age 5+ years; ≥1 dose refers to individuals receiving the first dose of the Pfizer and Moderna COVID-19 vaccines or a single shot of the J&J/Janssen vaccine; fully vaccinated refers to individuals fully vaccinated after receiving either both doses of the Pfizer and/or Moderna COVID-19 vaccines or a single shot of the J&J/Janssen vaccine.

		American Indian		Asian/Pacific Islander		Non-Hispanic Black		Hispanic		Non-Hispanic White		Total	
		N = 25,474		N = 35,503		N = 121,756		N = 151,188		N = 566,985		N = 900,906	
		n	%	n	%	n	%	n	%	n	%	n	%
Clinical													
	Case	2447	9.6	2576	7.3	9970	8.2	13,885	9.2	42,079	7.4	70,957	7.9
Vaccination ≥ 1 dose		8402	31.8	15,803	53.4	28,774	23.5	38,049	30.1	167,428	38.5	258,456	34.9
	Age group (years)												
	5–17	312	2.1	902	17.8	1063	3.7	2650	6.5	5570	8.6	10,497	7.2
	18–35	2054	21.5	4970	53.2	6159	16.7	13,657	32.8	36,670	33.0	63,510	30.7
	36–49	1980	34.6	3823	55.8	6116	27.7	9952	41.8	30,982	38.3	52,853	38.1
	50–64	2317	46.3	3711	69.8	8635	40.2	8144	56.2	40,672	44.5	63,479	46.2
	65+	1739	52.3	2397	78.8	6801	49.4	3646	66.6	53,534	61.9	68,117	60.9
	Gender												
	Female	4822	34.7	8469	52.9	16,574	25.7	20,695	34.0	90,890	40.7	141,450	37.3
	Male	3575	28.5	7290	53.7	12,158	20.9	17,300	26.4	76,315	36.1	116,638	32.3
Fully vaccinated		7424	28.1	13,505	45.6	24,156	19.7	30,681	24.3	151,111	34.8	226,877	30.7
	Age group (years)												
	5–17	109	2.1	378	7.5	337	1.2	1061	2.6	1977	3.0	3862	2.7
	18–35	1793	21.5	4369	46.8	4876	13.2	10,916	26.2	32,287	29.1	54,241	26.2
	36–49	1782	34.6	3345	48.8	5175	23.5	8255	34.7	28,429	35.2	46,986	33.9
	50–64	2112	46.3	3255	61.3	7625	35.5	7112	49.1	37,612	41.2	57,716	42.0
	65+	1628	52.3	2158	71.0	6143	44.6	3337	61.0	50,806	58.8	64,072	57.3
	Gender												
	Female	4292	30.9	7310	45.7	14,015	21.7	16,838	27.7	82,754	37.0	125,209	33.0
	Male	3130	25.0	6169	45.4	10,111	17.4	13,805	21.1	68,183	32.3	101,398	28.0

**Table 2 ijerph-19-08571-t002:** Distribution of reported demographic characteristics among reported COVID-19 cases by race/ethnicity in Oklahoma City–County, 12 March 2020–31 May 2021.

		American Indian N = 2447		Asian/Pacific Islander N = 2576		Non-Hispanic Black N = 9970		Hispanic N = 13,885		Non-Hispanic White N = 42,079		Total N = 70,957	
		n	%	n	%	n	%	n	%	n	%	n	%
Hospitalization *		84	3.4	74	2.9	608	6.1	509	5.3	1931	4.6	3206	4.5
Death **		28	1.1	19	0.7	187	1.9	139	1.0	946	2.2	1319	1.9
Sex													
	Male	1046	42.7	1222	47.4	4416	44.3	6096	43.9	19,485	46.3	32,265	45.5
	Female	1401	57.3	1354	52.6	5554	55.7	7789	56.1	22,594	53.7	38,692	54.5
Age group (years)													
	0–4	45	1.8	32	1.2	157	1.6	343	2.5	444	1.1	1021	1.4
	5–17	337	13.8	307	11.9	1191	11.9	2532	18.2	4227	10.0	8594	12.1
	18–35	849	34.7	894	34.7	3571	35.8	5534	39.9	12,969	30.8	23,817	33.6
	36–49	606	24.8	592	23.0	2245	22.5	3183	22.9	9095	21.6	15,721	22.2
	50–64	442	18.1	510	19.8	1801	18.1	1724	12.4	8483	20.2	12,960	18.3
	65+	168	6.9	241	9.4	1005	10.1	567	4.1	6859	16.3	8840	12.5
Comorbidities													
	≥1 comorbidity	418	17.1	238	9.2	1696	17.0	1549	11.2	6853	16.3	10,754	15.2
	Acute respiratory distress syndrome	9	0.4	7	0.3	70	0.7	68	0.5	357	0.8	511	0.7
	Chronic heart or circulatory disease	167	6.8	99	3.8	828	8.3	504	3.6	3209	7.6	4807	6.8
	Chronic liver disease	26	1.1	15	0.6	30	0.3	50	0.4	257	0.6	378	0.5
	Chronic lung disease	124	5.1	62	2.4	556	5.6	347	2.5	2207	5.2	3296	4.6
	Chronic renal failure	23	0.9	15	0.6	151	1.5	90	0.6	482	1.1	761	1.0
	Diabetes	172	7.0	93	3.6	573	5.7	697	5.0	1941	4.6	3476	4.9
	Disseminated intravascular coagulopathy	9	0.4	4	0.2	43	0.4	31	0.2	186	0.4	273	0.4
	Pneumonia	89	3.6	45	1.7	333	3.3	306	2.2	1995	4.7	2768	3.9

* 251 COVID-19-related hospitalizations where race/ethnicity was unknown. ** 184 COVID-19-related deaths where race/ethnicity was unknown.

**Table 3 ijerph-19-08571-t003:** Crude and adjusted * odds ratios (95% confidence intervals) between race/ethnicity and COVID-19-related outcomes (case, hospitalization, and death).

Race/Ethnicity	Case		Hospitalization				Death			
	OR_crude_	95% CI	OR_crude_	95% CI	OR_adj_ *	95% CI	OR_crude_	95% CI	OR_adj_ *	95% CI
Non-Hispanic White	Ref	-	Ref	-	Ref	-	Ref	-	Ref	-
American Indian	1.3	1.3–1.4	0.6	0.4–0.9	0.8	0.5–1.2	0.48	0.2–1.06	1.3	0.4–3.0
Asian/Pacific Islander	0.9	0.9–1.01	1.2	0.7–2.2	1.8	0.9–3.4	0.33	0.08–0.9	0.5	0.1–1.3
Non-Hispanic Black	1.1	1.08–1.13	0.9	0.7–1.1	1.4	1.1–1.8	1.01	0.7–1.4	1.7	1.2–2.5
Hispanic	1.3	1.2–1.3	0.6	0.5–0.7	1.2	0.9–1.5	0.33	0.2–0.5	1.2	0.7–1.8

* Adjusted for sex, age, and presence of at least one comorbidity.

## Data Availability

The corresponding author has full access to all of the data in the study and takes responsibility for the integrity of the data and accuracy of the data analyses. The data presented in this study are available on reasonable request from the corresponding author. The data are not publicly available because we are treating them consistently with all other public health disease surveillance data managed by the Oklahoma City County Health Department.

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
