# Peer review of "The Epidemiology of COVID-19 by Race/Ethnicity in Oklahoma City–County, Oklahoma (12 March 2020–31 May 2021)"

_ijerph, 2022, doi:10.3390/ijerph19148571_

Round 1
Reviewer 1 Report
ijerph-1783107
This is an interesting piece of research about the local experience in Tulsa, OK of COVID-19. The paper needs a lot of work but the data, and for the most part, the analysis appears adequate.
These are the major issues:
The paper lacks an adequate introduction. This should include what the research is about, why it is important and what issues will be addressed. Research questions would be helpful here. It is not necessary to tell the readers where Oklahoma is but a section in the review on the sociodemographic breakdown comparing the US, State and City (maybe an expansion of 1a?) might be helpful.
There does not appear to be an adequate review of the literature. Health disparity is a major literature in public health and it’s difficult to see how this informed the study. The Annual review of Public Health has some nice review articles that can help you build an authoritative review quickly. A lot of the literature in the discussion should have been pre-viewed in the Review.
The methodology section is hampered by the lack of a review. It needs to be better organized.
Results: These are truly terrifying statistics. The presentation looks adequate. I would think that Tables 2 & 3 should have a combined column. How do nursing homes fit in here?
The discussion would be stronger if it related more tightly to the data.
The conclusion needs to be better developed. What actions ought to be considered?
Not for this paper, but since you already have the data, consider geocoding the health care facilities/resources and comparing hot spot and non-hot spot areas.
Good luck with your revisions
Author Response
1. The paper lacks an adequate introduction. This should include what the research is about, why it is important and what issues will be addressed. Research questions would be helpful here. It is not necessary to tell the readers where Oklahoma is but a section in the review on the sociodemographic breakdown comparing the US, State and City (maybe an expansion of 1a?) might be helpful.
RESPONSE: We removed the mention of Oklahoma’s location. We have edited the text and added the following research question (line 55). “Given the length of the COVID-19 pandemic and the differences in strategies that governments, communities, and individuals employed in response, it is uncertain the extent to which racial and ethnic communities were impacted by COVID-19 over the time. Hence, our primary research question was whether vulnerable populations were equally vulnerable during the course of the pandemic, or if being impacted early bestowed protection against COVID-19-related outcomes later in the pandemic.”
2. There does not appear to be an adequate review of the literature. Health disparity is a major literature in public health and it’s difficult to see how this informed the study. The Annual review of Public Health has some nice review articles that can help you build an authoritative review quickly. A lot of the literature in the discussion should have been pre-viewed in the Review.
RESPONSE: We added more detail about the impact of health disparities and racial inequalities, along with a number of references to both the Introduction and Discussion sections. These include articles from the Annual Review of Public Health, as suggested.
3. The methodology section is hampered by the lack of a review. It needs to be better organized.
RESPONSE: We added subheadings and some additional details, and we hope that they addressed this concern.
4. Results: These are truly terrifying statistics. The presentation looks adequate. I would think that Tables 2 & 3 should have a combined column. How do nursing homes fit in here?
RESPONSE: Given the shared structure of Tables 1 and 2, and that it is unclear how a combined column would apply to Table 3, we believe what was requested was a combined column for Tables 1 and 2, and we have added the combined column to each table. At line 122, we have also added the following in response to nursing homes “In long-term care facilities, there were 1724 (1.7%) COVID-19 infection cases, 238 (6.9%) COVID-19-related hospitalizations, and 266 (17.7%) COVID-19-related deaths.” Unfortunately, we do not have these data stratified by race/ethnicity.
5. The discussion would be stronger if it related more tightly to the data.
RESPONSE: We made a number of edits to the discussion in which we elaborate on the reported data. For example, on lines 236-238, 250-255, and 275-288.
6. The conclusion needs to be better developed. What actions ought to be considered?
RESPONSE: We added some recommended actions to the Conclusion section (line 305) that reads, “We recommend increased integration of representatives from minority populations in public health-related initiatives. This includes inviting active participation from community members who are impacted by these initiatives, as well as recruiting and training community members to work in public health.”
7. Not for this paper, but since you already have the data, consider geocoding the health care facilities/resources and comparing hot spot and non-hot spot areas.
RESPONSE: Thank you for the suggestion. We look forward to exploring it.
Reviewer 2 Report
Abstract
Lines 21-22 - numbers for the rate should be provided except if authors were pressed for word count.
Introduction
Line 32 - what proportion of the total US population is Oklahoma? This information can be obtained from the table in Appendix 1 and stated here with the correct citation.
Lines 42-43 needs to be rephrased to read well like like 44-45 does.
Before paragraph in line 42, a sentence on availability of vaccine as effective intervention to curb the burden of COVID needs to be stated before the vaccination coverage.
Methods
It will be most suitable to have this section structured in state study design, data sources, power estimations, variables of interest including outcome and explanatory, analysis.
Discusssion
Citation or data source for Lines 227 -228 needs to be stated.
Conclusion
Lines 240-151... the authors provided relationship between SES and COVID hotspots... so another recommendation will be SES empowerment particulary for the population within the identified hotspot.
Author Response
1. Lines 21-22 - numbers for the rate should be provided except if authors were pressed for word count.
RESPONSE: We have added the rates as requested.
2. Line 32 - what proportion of the total US population is Oklahoma? This information can be obtained from the table in Appendix 1 and stated here with the correct citation.
RESPONSE: We edited that sentence (line 31) to read, “Oklahoma comprises 1.2% of the United States (US) population [1], and the distribution of race and ethnicity in Oklahoma County nearly mirrors the US (Table A1).”
3. Lines 42-43 needs to be rephrased to read well like 44-45 does.
RESPONSE: We edited the sentence (line 44) to read, “As of early June 2021, Oklahoma had administered just under 3 000 000 COVID-19 vaccine doses and had fully vaccinated over 1 336 000 people (approximately 34% of the state) [4].”
4. Before paragraph in line 42, a sentence on availability of vaccine as effective intervention to curb the burden of COVID needs to be stated before the vaccination coverage.
RESPONSE: We edited the sentence to read, “The COVID-19 vaccine was first available in Oklahoma County on December 14, 2020.”
5. It will be most suitable to have this section structured in state study design, data sources, power estimations, variables of interest including outcome and explanatory, analysis.
RESPONSE: We added the requested structure. Please note that because this was a descriptive analysis (as opposed to hypothesis testing) and we used all of the available surveillance data, we did not have any power calculations.
6. Citation or data source for Lines 227 -228 needs to be stated.
RESPONSE: We added the reference (lines 268-272).
7. Lines 240-151... the authors provided relationship between SES and COVID hotspots... so another recommendation will be SES empowerment particularly for the population within the identified hotspot.
RESPONSE: We added the suggested recommendation by editing the sentence (line 275) to read, “These hot spot analyses underscore the need to empower residents in the identified hot spots by increasing health literacy and providing additional public health infrastructure to improve equity in overall health among minority and vulnerable populations.”
Reviewer 3 Report
Abstract Aim does not match paper aim. Not once was vaccine mentioned in abstract, yet in introduction you clearly write about the vaccine being part of your aim. The vaccine uptake is also shown in tables and mentioned in discussion.
Abstract: We aimed to better understand the race/ethnic-specific COVID-19–related outcomes, with 11 respect to time, to respond more effectively to emerging variants. We summarized COVID-19 sur-12 veillance data from Oklahoma City-County during March 12, 2020-May 31, 2021. The distribution 13 of COVID-19 cases, hospitalizations, and deaths were summarized by racial/ethnic group and ZIP 14 code. We estimated racial/ethnic-specific disease odds ratios (OR) and 95% confidence intervals (CI) 15 adjusting for sex, age, and the presence of at least one comorbidity. Hot spot analysis was performed 16 using normalized values of cases, hospitalizations, and deaths generated from incidence rates per 17 100 000 population. During
vs. aim mentions in your introduction
We aimed to examine the epidemiological profile of 55 reported cases, hospitalizations, and deaths from COVID-19 by race/ethnicity. This in-56 cluded examining the geographical hot spots of these outcomes by race/ethnicity. Further, 57 we sought to examine the impact that vaccination had on these trends.
Your label Hispanics is problematic. Latinx is used in 2022.
Hispanic and Latino are often used interchangeably though they actually mean two different things. Hispanic refers to people who speak Spanish or are descended from Spanish-speaking populations, while Latino refers to people who are from or descended from people from Latin America.
https://www.yesprep.org/news/blog/featured/~board/blog/post/hispanic-vs-latinos-vs-latinx-explained
In line 74 you use Native American, in your chart you use American Indian. Native American is preferred throughout text.
If white is your reference group move white group to the start of each table.
Figure 1 shows 7-day hospital rate, but then there is a line about completed vaccine dose. Need to explain that line—which group is in the completed vaccine dose line? Is this everyone?
Further break down table 1 by age group and sex. Add to table 1 vaccination status by age group. It will be interesting to see who was receptive to vaccination not just by race, but also by age and sex.
Further break table 3 down by age. What are the death rates from different age groups and by sex? Where men dying more then women? Your analysis just by race does not tell a detailed story of what happened. Death rates might have been different by race for men vs women and for young vs old. Lumping everyone together does not tell us much new.
Note: In your discussion you report Blacks had lowest vaccination uptake. You should also note in your discussion that the uptake between Blacks & Latinx population was not that big of a difference, yet Blacks died at a much higher rate.
Line 203, COVID-19 is a pandemic not an epidemic.
Latinx health advantage not discussed
You state: our findings differ in that the Hispanic population 210 did not have a higher burden of COVID-19–related hospitalization and death
Here you should discuss reasons of why that could be. Latinx health advantage?? Not simply just that this is interesting for policy makers. You actually need to add a critical assessment.
No discussion on racism and bias in healthcare
Our findings from the COVID-19 hot spot 222 analysis revealed that hot spots of cases occurred in ZIP codes with high proportions of 223 Hispanics and non-Hispanic Whites, while hot spots of hospitalizations and deaths oc-224 occurred in ZIP codes with a high proportion of non-Hispanic Blacks. In this section you are going to need to do a more detailed discussion on racism in healthcare, medical doctors treatment of African Americans, and how African American people are more often gaslighted and are more likely to receive poor healthcare as compared to their white counterparts. As it stands now you have one line stating: These hot spot analyses underscore 233 the need to increase health literacy and provide additional public health infrastructure to 234 improve equity in overall health among minority and vulnerable populations. This is not a sufficient discussion of how African Americans experience the US health system and why they died at a higher rate. There are plenty of COVID-19 articles on stories of how African American COVID-19 patients were treated cruelly, not believed by doctors and so forth that you could use to better shape your discussion
Author Response
1. Abstract Aim does not match paper aim. Not once was vaccine mentioned in abstract, yet in introduction you clearly write about the vaccine being part of your aim. The vaccine uptake is also shown in tables and mentioned in discussion.
RESPONSE: We included our aim of describing vaccination status in the abstract and it now reads, “We aimed to better understand the racial-/ethnic-specific COVID-19–related outcomes, with respect to time, to respond more effectively to emerging variants. Surveillance data from Oklahoma City-County (March 12, 2020-May 31, 2021) was used to summarize COVID-19 cases, hospitalizations, deaths, and COVID-19 vaccination status by racial/ethnic group and ZIP code.”
2. Your label Hispanics is problematic. Latinx is used in 2022. Hispanic and Latino are often used interchangeably though they actually mean two different things. Hispanic refers to people who speak Spanish or are descended from Spanish-speaking populations, while Latino refers to people who are from or descended from people from Latin America.
RESPONSE: We agree with this important distinction. However, the variable from our surveillance data was labeled, “Hispanic.” Given that, we believe we should adhere to the variable label from the dataset to ensure accurate reporting of our results. To provide some explanation for using the term, “Hispanic,” we included a sentence (line 88) in the methods section to read, “The terms Hispanic and American Indian are used for the measure to accurately reflect those data captured in the surveillance. Hispanic is defined as those who are Spanish speakers and is inclusive of those who identify as Latinx.”
3. In line 74 you use Native American, in your chart you use American Indian. Native American is preferred throughout text.
RESPONSE: We agree that Native American is the preferred term. However, again, the label in the data is “American Indian.” It was our oversight to use “Native American” in line 74, and we have changed it to “American Indian” to be consistent with the dataset. We included a sentence (line 88-90) to read, “The terms Hispanic and American Indian are used for the measure to accurately reflect those data captured in the surveillance…American Indian is defined as those who are members of any of the indigenous peoples of America.”
4. If white is your reference group move white group to the start of each table.
RESPONSE: We moved the reference group as requested.
5. Figure 1 shows 7-day hospital rate, but then there is a line about completed vaccine dose. Need to explain that line—which group is in the completed vaccine dose line? Is this everyone?
RESPONSE: We clarified that line (line 165) in the figure. In the figure itself, it reads “Two Dose Vaccination start date” and in the header it reads, “The vertical dashed line indicated the first date by which an Oklahoma City-County resident could have received two COVID-19 vaccine doses.”
6. Further break down table 1 by age group and sex. Add to table 1 vaccination status by age group. It will be interesting to see who was receptive to vaccination not just by race, but also by age and sex.
RESPONSE: We added the requested data to Table 1.
7. Further break table 3 down by age. What are the death rates from different age groups and by sex? Where men dying more then women? Your analysis just by race does not tell a detailed story of what happened. Death rates might have been different by race for men vs women and for young vs old. Lumping everyone together does not tell us much new.
RESPONSE: We understand the primary concern is that the results in Table 3 could be confounded by age and sex. This is why we controlled for age and sex (and the presence of at least one comorbidity) in these adjusted analyses. Unfortunately, further stratifying the race-/ethnic-specific results by age results in small cell sizes and unstable estimates.
8. Note: In your discussion you report Blacks had lowest vaccination uptake. You should also note in your discussion that the uptake between Blacks & Latinx population was not that big of a difference, yet Blacks died at a much higher rate.
RESPONSE: We added a sentence (line 236) that reads, “An exception to this correlation is that the vaccination uptake among the Hispanic population was similar to the non-Hispanic Black population, but the Hispanic population experienced lower COVID-19-related death rates.”
9. Line 203, COVID-19 is a pandemic not an epidemic.
RESPONSE: We have edited to pandemic, as requested.
10. Latinx health advantage not discussed. You state: our findings differ in that the Hispanic population did not have a higher burden of COVID-19–related hospitalization and death. Here you should discuss reasons of why that could be. Latinx health advantage?? Not simply just that this is interesting for policy makers. You actually need to add a critical assessment.
RESPONSE: We edited that section (line 250) to read, “Although we do not have data to inform the reason for this difference, it has been reported that the Hispanic population in Oklahoma County has a lower mortality rate than for non-Hispanic populations [21]. These findings are informative for clinical providers, researchers, and policy makers so that results from studies in one location are not over-generalized to other populations that share the same racial/ethnic heritage.”
11. No discussion on racism and bias in healthcare. Our findings from the COVID-19 hot spot analysis revealed that hot spots of cases occurred in ZIP codes with high proportions of Hispanics and non-Hispanic Whites, while hot spots of hospitalizations and deaths occurred in ZIP codes with a high proportion of non-Hispanic Blacks. In this section you are going to need to do a more detailed discussion on racism in healthcare, medical doctors treatment of African Americans, and how African American people are more often gaslighted and are more likely to receive poor healthcare as compared to their white counterparts. As it stands now you have one line stating: These hot spot analyses underscore the need to increase health literacy and provide additional public health infrastructure to improve equity in overall health among minority and vulnerable populations. This is not a sufficient discussion of how African Americans experience the US health system and why they died at a higher rate. There are plenty of COVID-19 articles on stories of how African American COVID-19 patients were treated cruelly, not believed by doctors and so forth that you could use to better shape your discussion.
RESPONSE: We added the following paragraph (line 279) to the discussion to address this concern: “The factors contributing to health disparities and structural racism are complex and have historical roots [23]. An example of these disparities having long-term impacts on health is the finding that Black Americans receive poorer quality of perinatal and neonatal care, both of which are associated with increased chronic conditions later in life [24]. There is also a lack of representation of minorities in the field of public health [25]. Many of these factors can be addressed within the local context. Place-based, multisector, equity-oriented initiatives have been suggested as a strategy to address structural racism [23]. An example of a place-based strategy currently being employed in Oklahoma County includes hiring and embedding community health workers in community-based organizations that serve minority populations.”
Reviewer 4 Report
I have finished my review, I consider that the topic is interesting and the approach is clear in understanding aspects like inequities.
The concerns are related with the meaning of hospitalizations. I’m certain countries with restrictions for immigrants, being admitted at a hospital is more difficult because that have limited insurances or none. On the other had, also there is evidence that in certain regions, peaopme may have received benefits from home care and telemedicine. How was the medical insurances situation in this study, any information on the migratory status of hospitalized patients?
Please explain and you may include medical insurance as a variable as well.
Author Response
1. The concerns are related with the meaning of hospitalizations. I’m certain countries with restrictions for immigrants, being admitted at a hospital is more difficult because that have limited insurances or none. On the other hand, also there is evidence that in certain regions, people may have received benefits from home care and telemedicine. How was the medical insurances situation in this study, any information on the migratory status of hospitalized patients?
RESPONSE: Unfortunately, we do not have any data on the migratory status of hospitalized patients.
2. Please explain and you may include medical insurance as a variable as well.
RESPONSE: We have added a statement (line 296) to the Limitations section that the surveillance data we used does not include patient’s access to medical insurance.
Round 2
Reviewer 3 Report
Updated manuscript- approved.